∂ | **Open Peer Review** | Antimicrobial Chemotherapy | Research Article

# Antimicrobial activity of natural quinone-methide triterpenoids celastrol and pristimerin against pathogenic *Neisseria*

Alice Ascari,[1] Taha,[1] Rohan A. Davis,[1,2,3] Kate L. Seib,[1] Evgeny A. Semchenko[1]

**ABSTRACT** *Neisseria gonorrhoeae* and *Neisseria meningitidis* are closely related and important human pathogens that cause distinct diseases of clinical significance. While *N. meningitidis* infrequently demonstrates antibiotic resistance, the rapid emergence of multidrug-resistant *N. gonorrhoeae* isolates emphasizes the urgency of novel therapeutics against this pathogen. Recently, repurposing of approved drugs and the therapeutic potential of natural compounds have been explored as alternative strategies to treat gonococcal infections. We screened 54 natural bioactive compounds and identified celastrol and pristimerin, two quinone-methide triterpenoids, which exhibited bactericidal activity against *N. gonorrhoeae* 1291. The minimum inhibitory concentration (MIC) and minimum bactericidal concentration (MBC) for 10 susceptible and multidrug-resistant *N. gonorrhoeae* strains ranged between 0.4–11.3 mg/L and 0.7–23.3 mg/L, respectively. Both compounds were also bactericidal against a diverse panel of *N. meningitidis* strains representing multiple serogroups. The MIC and MBC for *N. meningitidis* strains ranged between 1.4–23.3 mg/L and 1.4–46.5 mg/L, respectively. Pristimerin effectively cleared *N. gonorrhoeae* infection in transformed primary cervical epithelial cells without inducing cytotoxicity at biologically active concentrations. In contrast, celastrol exhibited a modest cytotoxic effect on the cells. In summary, we screened a library of bioactive compounds and identified the natural quinone-methide triterpenoids celastrol and pristimerin, as antimicrobial leads against pathogenic *Neisseria* species. These compounds represent promising scaffolds for the future development of therapeutics targeting gonorrhea.

**IMPORTANCE** *Neisseria gonorrhoeae* and *Neisseria meningitidis* cause clinically significant diseases, but rising multidrug-resistant *N. gonorrhoeae* creates an urgent need for new therapies. We screened 54 natural bioactive compounds from the NatureBank library and identified celastrol and pristimerin, two quinone-methide triterpenoids with notable bactericidal activity against susceptible and resistant *N. gonorrhoeae*, as well as diverse *N. meningitidis* serogroups. Celastrol showed stronger activity but limited selectivity and known toxicity, indicating a need for structural optimization. Pristimerin exhibited a favorable safety margin while maintaining antimicrobial potency. These findings highlight natural triterpenoids as promising scaffolds for lead optimization and development into novel antimicrobials to combat antibiotic-resistant gonorrhea.

**KEYWORDS** *Neisseria gonorrhoeae*, *Neisseria meningitidis*, natural compounds, NatureBank, antibacterial drug screening, antimicrobial activity, cytotoxicity, quinone-methide, celastrol, pristimerin

T he pathogenic *Neisseria* species, *N. gonorrhoeae* (Ng) and *N. meningitidis* (Nm), are closely related gram-negative bacteria with high genetic similarity but distinct disease profiles (1–3). Over 82 million gonorrhea cases occur annually (1). While many Ng infections remain asymptomatic, particularly in women (up to 80%) (4, 5), untreated

**Peer Reviewer** William M. Shafer, Emory University School of Medicine, Atlanta, Georgia, USA

Address correspondence to Evgeny A. Semchenko, e.semchenko@griffith.edu.au.

Kate L. Seib and Evgeny A. Semchenko contributed equally to this article.

The authors declare no conflict of interest.

See the funding table on p. 10.

cases can result in pelvic inflammatory disease, infertility, and adverse pregnancy outcomes, while in men, symptomatic urethritis can progress to epididymo-orchitis, urethral strictures, and infertility (5, 6). In contrast, Nm causes invasive meningococcal disease (IMD), manifesting as meningitis and septicemia. Although asymptomatic nasopharyngeal carriage is common, Nm can breach mucosal barriers and disseminate systemically, leading to rapid, severe disease (2). Globally, Nm causes over 1.2 million IMD cases annually, with a 10%–15% fatality rate and long-term sequelae in 20% of survivors (2, 7).

Ng and Nm pose significant public health challenges, exacerbated by rising antimicrobial resistance (AMR) (8). Notably, the WHO (9), CDC (8), and the Australian National AMR Strategy (10) list Ng as a priority pathogen for new therapeutic development, citing concerns that it may become untreatable. Current treatment options are limited, with ceftriaxone monotherapy the standard Ng regimen, and ceftriaxone (or high-dose penicillin G, where susceptible) used for Nm infections (7, 8). The absence of a Ng vaccine and incomplete global Nm vaccine coverage, alongside the emergence of non-vaccine serogroups, further hinder control efforts and underscore the need for novel therapeutics (11).

To address rising AMR, novel therapeutics have been trialed, particularly against Ng. Strategies include non-toxic, cell-penetrating peptides that inhibit bacterial invasion (12) and cytokine responses, and small-molecule inhibitors targeting essential enzymes such as LpxC, crucial for lipid A biosynthesis and membrane integrity (13). Chemically modified novel bacterial topoisomerase inhibitors, including gepotidacin (14) and zoliflodacin (15), have shown efficacy in phase III trials, with reduced resistance emergence due to dual inhibition of DNA gyrase and topoisomerase IV (16). Furthermore, drug repurposing offers a cost-effective, accelerated approach, leveraging the known safety and pharmacokinetics of approved drugs (17–20). Agents such as methyldopa (17) and carbamazepine (18), originally developed for hypertension and epilepsy, respectively, have cleared multidrug-resistant (MDR) Ng infection in cervical cell models. Anthranilic acid derivatives (tolfenamic, flufenamic, and meclofenamic acids) also exhibit potent intracellular activity, reducing Ng load and associated cytokine production by over 99% in infected endocervical cells (19). These strategies reflect growing interest and progress in therapeutic development for Ng, yet its capacity for resistance and the absence of a vaccine underscore the ongoing need for novel interventions.

Natural compounds, characterized by lower toxicity and fewer adverse effects than synthetic agents, have demonstrated therapeutic potential across infections, cancer, and inflammatory conditions (21). Given the limited prevention and treatment options for gonococcal infections, investigating their bioactivity against Ng is warranted. In this study, we screened natural, bioactive compounds from Australian plants and animals (NatureBank) (22) to identify novel antimicrobial agents Ng and Nm.

## MATERIALS AND METHODS

### Strains and growth conditions

*N. gonorrhoeae* strains 1291, FA1090, WHO H, WHO K, WHO Q, WHO R, WHO S2, WHO X, WHO Z, and a WHO X *mtrE* knockout were grown on GC agar plates (Oxoid) or in GC broth supplemented with IsoVitaleX (Becton Dickinson) (23, 24). *N. meningitidis* strains M01 240070, M03 241125, M99 243594, L94 5016, MC58, and MC58 ¢3 were grown on GC agar plates or in GC broth for 16 hours at 37°C with 5% $CO_2$ (25, 26).

### Compound library screening

A library comprising 54 high-quality, chemically defined natural compounds was obtained from NatureBank (Dataset S1) (22). Two microliters of 5 mM test compounds was dispensed in duplicate into clear-bottom 96-well plates (Greiner) containing 48 µL

of GC broth per well. Control wells contained media alone and media supplemented with a matching concentration of DMSO (2%), which did not have any visible impact on bacterial growth. The antibiotic azithromycin (Merck) was included in separate wells at a final concentration of 50 mg/L and served as a positive control for growth inhibition. Each well was then inoculated with 50 µL of Ng 1291 suspension, adjusted to an optical density at 600 nm ($OD_{600}$) of 0.01 in GC broth resulting in a final compound concentration of 100 µM. Plates were incubated at 37°C with 5% $CO_2$ for 20 hours. Following incubation, bacterial growth was visually assessed and scored according to Clinical Laboratory Standards Institute (CLSI) guidelines as described previously (23, 27).

## Extraction and purification of celastrol and pristimerin

The extraction and purification of the two most active compounds, celastrol and pristimerin, from the roots of the Queensland rainforest plant *Maytenus bilocularis* have been reported previously (28). Purity of natural products celastrol and pristimerin was determined to be >95% following UHPLC-MS analysis (Supplementary Data). Compound stock solutions were prepared in DMSO to 5 mg/L.

## Antibiotic and compound susceptibility testing

The minimum inhibitory concentration (MIC) and minimum bactericidal concentration (MBC) of compounds were determined using a microdilution method according to the CLSI guidelines, as described previously, for efficient, parallel MIC/MBC testing across strains, and to match subsequent assay format (23, 27). Test compounds were prepared in a series of twofold serial dilutions in 50 µL of GC broth. Control wells contained GC broth with either 50 mg/L azithromycin or no added compound. All wells, including azithromycin and no-compound controls, contained matched DMSO (≤0.5% vol/vol), which had no effect on bacterial growth or viability. Each well was inoculated with 50 µL of Ng suspension, adjusted to an $OD_{600}$ of 0.01 in GC broth. Plates were incubated at 37°C with 5% $CO_2$ for 20 hours. The MIC was defined as the lowest concentration of compound at which no visible bacterial growth was observed. For MBC determination, assay plates were removed from the incubator after 45 minutes of exposure and diluted 1:10 in GC broth. Five microliters of diluted samples was spot plated on GC agar plates and incubated at 37°C with 5% $CO_2$ for 20 hours. The MBC was defined as the lowest concentration of antibiotic/compound at which no colony growth was observed on the agar plates. MIC and MBC group means were compared using Student's *t*-test on $log_2$-transformed data with 95% confidence intervals (CIs) reported. Correlation analysis was performed using a simple linear regression model in GraphPad Prism (v10).

## Cervical cell infection studies

Cell infection assays were performed with E6/E7 transformed primary cervical epithelial (tCX) cells as described previously (29, 30). Briefly, cells were grown as monolayers in 96-well cell culture plates to complete confluence in keratinocyte serum-free medium (KSFM) supplemented with epidermal growth factor and bovine pituitary extract (Gibco). Ng WHO X inoculum was prepared from mid-log cultures to an $OD_{600}$ of 0.1 in KSFM. Prior to infection, tCX cell monolayers were washed with 100 µL of HBSS (Gibco). Assay wells were inoculated with 50 µL of Ng suspension (~$10^7$ colony-forming units [CFU]/mL, multiplicity of infection: 10) and plates incubated for 1 hour at 37°C with 5% $CO_2$. Non-adherent bacteria were removed by washing cell monolayers with 100 µL of HBSS prior to adding 100 µL of compounds serially diluted in KSFM. Assay plates were incubated for 20 hours at 37°C with 5% $CO_2$. Wells were washed three times with HBSS, and cells were lysed with 1% saponin (in GC broth). After 10 minutes of incubation, the well contents were vigorously resuspended, serially diluted in broth, and plated on GC agar plates. Bacterial CFUs were enumerated the following day, and percent survival from triplicate wells was calculated relative to the no-treatment control. Compound 50% inhibitory concentration ($IC_{50}$) and 90% inhibitory concentration ($IC_{90}$) values

were estimated from percent survival data using nonlinear regression (four-parameter logistic model) in GraphPad Prism (v10). Statistical comparisons between groups were performed using the extra sum-of-squares $F$ test.

## Cell cytotoxicity and selectivity index assessment

Compound cytotoxicity was assessed in cervical epithelial cells as previously described (31). Briefly, tCX were cultured as outlined above. Cell monolayers were washed with 100 µL of HBSS (Gibco) and incubated with serial dilutions of test compounds at 37°C in 5% $CO_2$ for 16 hours. Resazurin was then added to each well at a final concentration of 10 µM. Controls included untreated cells (live control) and cells treated with 0.1% (vol/vol) Triton X-100 (dead control). Plates were incubated for an additional 4 hours at 37°C in 5% $CO_2$, and fluorescence was measured at 530/590 nm using a TECAN Infinite 200 Pro plate reader with a pre-optimized gain setting. Cell viability was calculated as the percentage of fluorescence relative to no-treatment control wells. Each condition was tested in triplicate. Compound 50% cytotoxic concentration ($CC_{50}$) values were estimated from percent viability data using nonlinear regression (four-parameter logistic model) in GraphPad Prism (v10). Statistical comparisons between groups were performed using the extra sum-of-squares $F$ test. The selectivity index (SI) was calculated as the ratio of the $CC_{50}$ to the $IC_{50}$, using the formula: $SI = CC_{50}/IC_{50}$ (32).

## Physicochemical properties of the compounds

Key physicochemical descriptors of celastrol and pristimerin were calculated using SwissADME (http://www.swissadme.ch) (33).

## RESULTS

### Screening for compounds that inhibit bacterial growth

To identify novel antimicrobial agents, we screened a library of compounds from NatureBank using a high-throughput growth inhibition assay against *N. gonorrhoeae* strain 1291. Each compound was tested at a single concentration of 100 µM. Of the 54 compounds screened, two structurally related compounds, celastrol and pristimerin, inhibited bacterial growth and were selected for further investigation (Fig. 1). The potency criterion used for selection was based on complete inhibition of visible bacterial growth at the tested concentration, as determined by visual scoring in accordance with CLSI guidelines.

### Antibacterial activity of hit compounds against pathogenic *Neisseria*

The antibacterial potential of celastrol and pristimerin was evaluated against a panel of diverse Ng strains. Both celastrol and pristimerin displayed bactericidal activity against susceptible and MDR Ng strains, with the MIC ranging between 0.4–5.6 mg/L and 1.5–11.6 mg/L and the MBC ranging between 0.7–11.3 mg/L and 1.5–23.3 mg/L, respectively (Table 1). The MIC of celastrol was, on average, three times lower than that of pristimerin, based on geometric mean MICs across 10 Ng strains (celastrol = 1.32 mg/L [95% CI: 0.74–2.36] vs. pristimerin = 4.41 mg/L [95% CI: 2.74–7.09], $P = 0.002$). The MBC of celastrol was also significantly lower than that of pristimerin (geometric mean MBC: celastrol = 5.62 mg/L [95% CI: 3.16–9.97] vs. pristimerin = 8.21 mg/L [95% CI: 5.78–11.66], $P = 0.012$). Additionally, the resistance-nodulation-division type efflux pump outer membrane channel MtrE mutant strain WHO X Δ*mtrE* was 2–4 times more susceptible to both compounds than the WHO X wild-type strain (Table 1).

Similar to their activity against Ng, celastrol and pristimerin exhibited bactericidal activity against a diverse panel of Nm strains, including serogroups A, B, C, Y, and W135 strains. The MIC for celastrol and pristimerin ranged between 1.4–2.8 mg/L and 2.9–23.3 mg/L, and the MBC ranged between 1.4–22.6 mg/L and 5.8–46.5 mg/L, respectively (Table 2). The MIC of celastrol was, on average, five times higher than that of pristimerin,

**FIG 1** Chemical structures for celastrol and pristimerin.

based on geometric mean MICs across five strains (celastrol = 2.44 mg/L [95% CI: 1.66–3.58] vs. pristimerin = 11.63 mg/L [95% CI: 3.43–39.39], $P$ = 0.008). The MBC of celastrol was not significantly different from that of pristimerin (geometric mean MBC: celastrol = 3.70 mg/L [95% CI: 0.87–15.69] vs. pristimerin = 10.10 mg/L [95% CI: 3.29–31.07], $P$ = 0.18). The antimicrobial activity of celastrol was not affected by Nm capsule expression or serogroup. However, the MIC of pristimerin was fourfold lower for non-encapsulated MC58 ¢3 (5.8 mg/L) compared to encapsulated MC58 (23.3 mg/L).

**TABLE 1** MIC and MBC of celastrol and pristimerin in a panel of *N. gonorrhoeae* strains[a]

| Strain | Isolated | Country | Azithromycin sensitivity[b] | Ceftriaxone sensitivity[c] | Celastrol | | Pristimerin | |
|---|---|---|---|---|---|---|---|---|
| | | | | | MIC | MBC | MIC | MBC |
| 1291 | 1974 | USA | S[d] | S[d] | 2.8 | 0.7 | 2.9 | 5.8 |
| FA1090 | 1983 | USA | S[e] | S[e] | 0.7 | 5.6 | 2.9 | 5.8 |
| WHO F | 1991 | Canada | S[f] | S[f] | 0.4 | 5.6 | 2.9 | 5.8 |
| WHO H | 2011 | Austria | S[f] | LLR[f] | 1.4 | 11.3 | 5.8 | 5.8 |
| WHO K | 2003 | Japan | S[f] | S[f] | 0.7 | 5.6 | 1.5 | 11.6 |
| WHO Q | 2018 | UK | HLR[f] | R[f] | 1.4 | 5.6 | 5.8 | 11.6 |
| WHO R | 2015 | Japan | S[f] | R[f] | 5.6 | 11.3 | 11.6 | 11.6 |
| WHO S2 | 2020 | Sweden | R[f] | S[f] | 2.8 | 5.6 | 11.6 | 23.3 |
| WHO Z | 2013 | Australia | S[f] | R[e] | 0.7 | 11.3 | 2.9 | 5.8 |
| WHO X | 2009 | Japan | S[f] | HLR[f] | 1.4 | 5.6 | 5.8 | 5.8 |
| WHO X ΔmtrE | –[g] | – | S[e] | R[e] | 0.7 | 2.8 | 1.5 | 1.5 |

[a]Antimicrobial sensitivity: S, susceptible; R, resistant; LLR, low level resistance; HLR, high-level resistance.
[b]Azithromycin sensitivity as determined by epidemiological cut-off of MIC > 1 mg/L as no clinical breakpoints are reported by EUCAST.
[c]Ceftriaxone sensitivity as per EUCAST guidelines (www.eucast.org/clinical_breakpoints).
[d]Determined in this study (Table S1).
[e]Reported in Evert et al. (23).
[f]Reported in Unemo et al. (24).
[g]"–" isolation year and origin are not applicable; WHO X ΔmtrE is a laboratory-derived mutant of WHO X.

**TABLE 2** MIC and MBC of celastrol and pristimerin in a panel of *N. meningitidis* strains

| Strain | Serogroup | Capsular polysaccharide repeating unit | Celastrol | | Pristimerin | |
|---|---|---|---|---|---|---|
| | | | MIC | MBC | MIC | MBC |
| M99 243594 | A | $[\rightarrow6)\text{-}\alpha\text{-D-ManNAc}(3/4\text{-OAc})\text{-}(1\rightarrow OPO_3^-)]_n$ | 2.8 | 22.6 | 23.3 | 46.5 |
| MC58 | B | $[\rightarrow8)\text{-}\alpha\text{-D-Neu5Ac-}(2\rightarrow)]_n$ | 2.8 | 5.6 | 23.3 | 11.6 |
| MC58 ¢3 | B | $-^a$ | 2.8 | 5.6 | 5.8 | 11.6 |
| L94 5016 | C | $[\rightarrow9)\text{-}\alpha\text{-D-Neu5Ac}(7/8\text{-OAc})\text{-}(2\rightarrow)]_n$ | 2.8 | 2.8 | 5.8 | 5.8 |
| M01 240070 | W135 | $[\rightarrow6)\text{-}\alpha\text{-D-Gal-}(1\rightarrow4)\text{-}\alpha\text{-D-Neu5Ac}(7/9\text{-OAc})\text{-}(2\rightarrow)]_n$ | 1.4 | 1.4 | 2.9 | 5.8 |
| M03 241125 | Y | $[\rightarrow6)\text{-}\alpha\text{-D-Glc-}(1\rightarrow4)\text{-}\alpha\text{-D-Neu5Ac}(7/9\text{-OAc})\text{-}(2\rightarrow)]_n$ | 2.8 | 1.4 | 23.3 | 5.8 |

*a* "–" not applicable; MC58 ¢3 is an acapsular derivative strain of MC58.

## *In vitro* efficacy of compounds against cervical cell infection by *N. gonorrhoeae*

To assess the therapeutic potential of celastrol and pristimerin against an established Ng infection, *in vitro* infection assays were performed using cervical epithelial cells. Both compounds demonstrated significant antimicrobial activity against the MDR Ng strain WHO X (Fig. 2a). Dose-response analyses were performed to estimate the concentrations required to inhibit 50% and 90% of bacterial viability ($IC_{50}$ and $IC_{90}$, respectively). Celastrol exhibited greater potency against Ng WHO X, with $IC_{50}$ and $IC_{90}$ values of 2.26 mg/L (95% CI: 2.03–2.49) and 3.26 mg/L (95% CI: 2.49–4.02), respectively, compared to pristimerin, which showed $IC_{50}$ and $IC_{90}$ values of 4.43 mg/L (95% CI: 3.61–5.25) and 6.12 mg/L (95% CI: 1.09–11.15), respectively (*F* test, *P* < 0.0001).

### Evaluation of cytotoxicity and therapeutic selectivity

Cytotoxicity of the compounds was evaluated in cervical epithelial cells using a resazurin-based viability assay. Celastrol exhibited dose-dependent cytotoxicity, with a $CC_{50}$ of 8.09 ± 0.29 mg/L (95% CI: 6.84–9.34), and notable loss of cell viability observed at concentrations exceeding 3 mg/L (Fig. 2b). In contrast, pristimerin showed substantially lower cytotoxicity, with no significant impact on cell viability up to 23 mg/L and an estimated $CC_{50}$ of approximately 258 mg/L. To assess the therapeutic window of each compound, the SI was calculated as the ratio of the $CC_{50}$ in cervical epithelial cells to the $IC_{50}$ against Ng WHO X. Celastrol exhibited a relatively low SI of 3.58 ± 0.15 (95% CI: 2.93–4.23), indicating limited selectivity between host cell toxicity and antibacterial activity. In contrast, pristimerin demonstrated a substantially higher SI of approximately

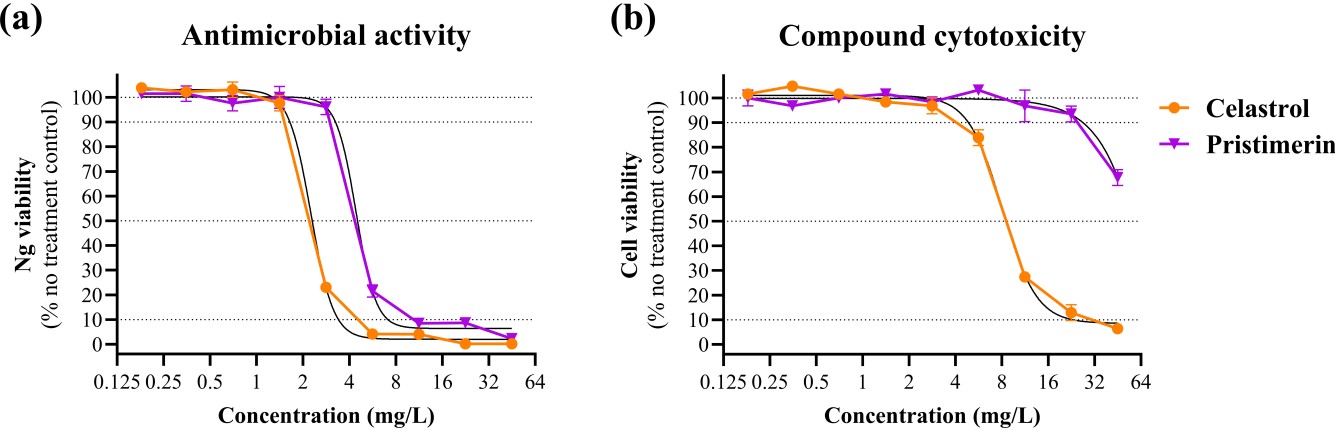

**FIG 2** *In vitro* antimicrobial activity (a) and cytotoxicity (b) of celastrol and pristimerin in a *N. gonorrhoeae* cervical cell infection model. Cervical epithelial cell (tCX) monolayers were infected with *N. gonorrhoeae* WHO X and treated with celastrol or pristimerin. (a) Bacterial viability is shown as percentage of recovered *N. gonorrhoeae* colony forming units relative to no treatment control ± SEM. (b) Cytotoxicity of celastrol and pristimerin in uninfected tCX cells was assessed using a resazurin-based viability assay. Cervical cell viability is expressed as a percentage relative to the no-treatment control. Dose-response curves were fitted using nonlinear regression (four-parameter logistic model) and are represented by black lines on the graph.

**TABLE 3** Physicochemical properties of celastrol and pristimerin

| Physicochemical properties | Celastrol | Pristimerin |
|---|---|---|
| Molecular formula | $C_{29}H_{38}O_4$ | $C_{30}H_{40}O_4$ |
| Molecular weight | 450.6 g/mol | 464.6 g/mol |
| # rotatable bonds | 1 | 2 |
| # H-bond acceptors | 4 | 4 |
| # H-bond donors | 2 | 1 |
| Topological surface area | 74.60 Å² | 63.60 Å² |
| Consensus Log Po/w | 5.12 ± 1.37 | 5.58 ± 1.08 |
| Fraction Csp3 | 0.66 | 0.67 |
| Molar refractivity | 131.29 | 135.61 |

58. Although this value could not be precisely quantified due to the upper limit of the cytotoxicity assay, it exceeds the commonly applied threshold of 10 for preliminary selectivity in cell-based antibacterial discovery (34, 35), suggesting a favorable safety margin and supporting its potential for further development.

## Physicochemical properties of celastrol and pristimerin

Key physicochemical descriptors of celastrol and pristimerin were determined using SwissADME (33) to establish their suitability as lead compounds for further development (Table 3). Celastrol and pristimerin exhibit favorable properties for continued structure-activity relationship optimization in medicinal chemistry, including molecular weights under 500 Da, fraction of sp3-hybridized carbons (Fsp3) greater than 0.5, and fewer than 10 rotatable bonds (Fig. 1). Given the variability in reported LogP values across computational methods, a consensus LogP was calculated by averaging values from five commonly used algorithms (iLOGP, XLOGP3, WLOGP, MLOGP, and SILICOS-IT) (33). The resulting consensus LogP values were 5.12 ± 1.37 and 5.58 ± 1.08 for celastrol and pristimerin, respectively. While slightly above the typical threshold for drug-likeness (LogP ≤ 5), these values do not inherently preclude further development. Both compounds pass Lipinski's (36) (single violation; LogP > 5) and Veber's (37) (no violations) criteria for oral bioavailability (Table 3). Additionally, celastrol and pristimerin pass Brenk (38) and Pan Assay Interference Compounds PAINS (39) filters, indicating low risk of assay interference, supporting their potential as antibacterial lead compounds for further development. While promising, their redox-active cores could influence off-target interactions in certain biological contexts and should be further explored during lead optimization.

## DISCUSSION

Nature has historically been a cornerstone of drug discovery and continues to hold immense promise for identifying novel bioactive compounds. NatureBank, a unique drug discovery platform, harnesses this potential through its extensive library of natural product extracts (~21,000), fractions (~105,000), and compounds (~54) derived from Australian plants, fungi, and marine invertebrates (22). However, natural product extracts are complex mixtures that are not chemically defined, and their constituents are often present in very low concentrations, which could be below the threshold required to elicit a measurable biological activity. Thus, our approach to identifying novel antimicrobials targeting pathogenic *Neisseria* was to focus on a small library of high-quality chemically defined compounds. From the library of 54, we identified two compounds that inhibited the growth of *N. gonorrhoeae* 1291. Celastrol and its methyl ester analog, pristimerin, are quinone-methide triterpenoids derived from the roots of *Celastraceae* plants, such as *Tripterygium wilfordii* (40). Both compounds are known to exhibit a range of pharmacological activities, and various therapeutic uses have been proposed (40, 41), including antimicrobial effects (42, 43); however, their activity against pathogenic *Neisseria* species has not been described.

We subsequently showed that both celastrol and pristimerin exhibit antimicrobial activity, with MICs and MBCs indicating both bacteriostatic and bactericidal effects against diverse Ng and Nm strains. Previous studies with other microbes have shown that celastrol has greater antimicrobial efficacy than pristimerin (42, 44), which is consistent with our findings for both *Neisseria* species. The geometric mean MICs of celastrol for Ng and Nm were 1.32 mg/L and 2.44 mg/L, and the geometric mean MICs of pristimerin were 4.41 mg/L and 11.63 mg/L, respectively.

The antimicrobial activity of these compounds was reported to involve multiple mechanisms, including disruption of the cytoplasmic membrane and covalent modification of proteins containing reactive cysteine residues, potentially interfering with key metabolic processes such as DNA and RNA synthesis (42). This activity likely stems from the intrinsic electrophilic nature of celastrol and pristimerin, which enables them to form thioether adducts via Michael addition between their α,β-unsaturated carbonyl moieties and nucleophilic thiol groups of cysteine residues (45). Thus, it is likely that due to their broad-spectrum activity on multiple targets, there was no correlation between the effects of these compounds and the resistance profiles of MDR Ng to azithromycin and ceftriaxone, which target the 50S ribosomal subunit and penicillin-binding proteins, respectively (46). In contrast to several non-*Neisseria* gram-negative species where pristimerin showed no bactericidal activity at concentrations exceeding 400 mg/L (47), *Neisseria* species exhibited significantly lower MICs under our assay conditions, indicating greater susceptibility. This suggests that species-specific envelope or physiological features may influence pristimerin activity (48). Interestingly, the WHO X MtrE efflux pump mutant strain was more susceptible to both celastrol and pristimerin, suggesting that Ng may detoxify these compounds via MtrE-dependent efflux, which preferentially exports hydrophobic and amphiphilic molecules such as azithromycin (49, 50). Notably, celastrol and pristimerin exhibit pronounced hydrophobicity and possess planar, conjugated ring systems that may also facilitate interaction with the efflux regulator MtrR through π–π and general hydrophobic contacts, akin to steroid hormones such as progesterone and testosterone, which have been shown to bind MtrR and modulate *mtrCDE* expression (51). These shared physicochemical features suggest a potential intersection with Ng efflux regulation, warranting future investigation into their binding to MtrR and activity across isogenic transporter mutants (48, 51). While the efflux interaction remains hypothetical, celastrol and pristimerin nevertheless demonstrated robust activity against the WHO panel of Ng strains with extensive AMR profiles, including those with reduced susceptibility to ceftriaxone and high-level resistance to azithromycin, highlighting the therapeutic potential of triterpenoid scaffolds in the context of multidrug-resistant and extensively drug-resistant gonococcal infections.

There are 12 recognized Nm serogroups, defined by their capsular polysaccharide composition (52). Capsular polysaccharide structural differences may influence the activity of celastrol and pristimerin. For example, the Nm serogroup B capsule is composed of highly negatively charged polysialic acid, whereas the Nm serogroup C capsule is O-acetylated, which reduces its overall charge and hydrophilicity. To assess whether capsule expression affects antimicrobial activity, we tested strains from the five major disease-associated serogroups, A, B, C, Y, and W135. Antimicrobial activity of celastrol was largely unaffected by capsule expression or serogroup, with consistent MIC and MBC values across all strains and serogroups. In contrast, pristimerin showed enhanced potency in the absence of the capsule: the MIC for non-encapsulated MC58 ¢3 was fourfold lower than that for encapsulated MC58. However, this difference was not observed in the MBC assay. Together, these findings suggest that the antimicrobial activity of celastrol and pristimerin is not primarily determined by capsule type or serogroup. Instead, it is likely that the amount of capsule expressed, rather than its specific chemical composition, plays a more significant role in modulating drug susceptibility, particularly for hydrophobic compounds like pristimerin. This may be particularly relevant during infection, as Nm exhibits strain-dependent differences in capsule production and modulates capsule expression at distinct stages of infection (52).

We further assessed the therapeutic potential of celastrol and pristimerin against Ng using *in vitro* infection and cytotoxicity assays with cervical epithelial cells. Notably, higher bactericidal concentrations of compounds were required in the cervical cell infection model compared to planktonic MIC/MBC values. This reduced susceptibility is a well-established phenomenon in cell-based infection assays and likely stems from host-cell environmental factors, including bacterial adhesion, invasion, and intracellular localization, which are known to shift concentration-response curves. Celastrol exhibited greater potency than pristimerin against Ng WHO X ($IC_{90}$ 3.26 mg/L vs. 6.12 mg/L), but it was considerably more toxic to host cells. The $CC_{50}$ value of celastrol was 8.09 ± 0.29 mg/L, whereas the $CC_{50}$ for pristimerin was estimated at ~258 mg/L, though this could not be precisely determined due to limitations in the assay's detection range. Pristimerin has also been reported to exhibit variable cytotoxicity (e.g., $CC_{50}$ ~6 mg/L in Vero cells) (47), underscoring differences across cell types and assay conditions. Celastrol hence exhibited a relatively low SI of 3.58 ± 0.15, indicating limited selectivity between antibacterial activity and host cell toxicity (32). This is consistent with other reports identifying celastrol as a compound with notable off-target effects, including cardiotoxicity, hepatotoxicity, and neurotoxicity (53). Despite these limitations, numerous celastrol derivatives and conjugates have been developed to improve efficacy and reduce toxicity (54, 55). This, along with similar studies, reinforces the potential of triterpenoids as scaffolds for Ng drug development (56).

Collectively, our findings highlight the potential of celastrol and pristimerin as promising antibacterial scaffolds. While celastrol demonstrated stronger antimicrobial activity, its limited selectivity and known toxicity profile underpin the need for structural optimization. In contrast, pristimerin showed a more favorable safety margin and maintained effective antibacterial activity. Importantly, both compounds meet Lipinski's (36) and Veber's (37) criteria for oral bioavailability and are not flagged by Brenk (38) or PAINS (39) filters, supporting their potential as drug-like leads. Nonetheless, their redox-active cores merit further evaluation to rule out context-dependent off-target interactions. Taken together, these drug-like properties support their candidacy for lead optimization and development into novel antimicrobials, which are urgently needed in the fight against AMR in Ng.

## ACKNOWLEDGMENTS

We acknowledge the Benefit Sharing Agreement between the Queensland Government and Griffith University under the Biodiscovery Act 2004 (QLD), enabling equitable benefit sharing from biodiscovery using Queensland-sourced Native Biological Material. The authors thank NatureBank for the supply of the compound library screened during these studies, which were purified from various plant and marine invertebrate samples collected from the State of Queensland. The authors acknowledge the UK Health Security Agency for providing N. meningitidis strains.

This research was supported by the National Health and Medical Research Council of Australia (K.L.S. is supported by Investigator Grant GNT2017383) and the Australian Research Council for NMR and MS equipment grants (LE0668477, LE140100119, and LE0237908).

A.A.: data curation, investigation, writing—review and editing. T.: data curation, investigation, writing—review and editing. R.A.D.: conceptualization, funding acquisition, methodology, project administration, resources, visualization, writing—review and editing. K.L.S.: conceptualization, funding acquisition, project administration, resources, supervision, writing—review and editing. E.A.S.: conceptualization, data curation, formal analysis, investigation, methodology, project administration, resources, supervision, validation, visualization, writing—original draft, writing—review and editing.

The authors declare no conflicts of interest, financial or otherwise, relevant to the content of this article. The study was conducted with full transparency, and no external parties influenced the design, execution, analysis, or reporting of the research.

During the preparation of this work, the author(s) used Microsoft Copilot (Griffith University enterprise version) in order to identify improvements in writing style, including grammar and phrasing. After using this tool/service, the author(s) reviewed and edited the content as needed and take(s) full responsibility for the content of the publication.

## AUTHOR AFFILIATIONS

[1]Institute for Biomedicine and Glycomics, Griffith University, Gold Coast, Queensland, Australia
[2]School of Environment and Science, Griffith University, Nathan, Queensland, Australia
[3]NatureBank, Griffith University, Nathan, Queensland, Australia

## AUTHOR ORCIDs

Alice Ascari https://orcid.org/0000-0002-5626-7973
Rohan A. Davis http://orcid.org/0000-0003-4291-7573
Kate L. Seib http://orcid.org/0000-0002-7094-3528
Evgeny A. Semchenko http://orcid.org/0000-0002-5188-7946

## FUNDING

| Funder | Grant(s) | Author(s) |
| --- | --- | --- |
| Australian Research Council | LE0237908 | Rohan A. Davis |
| Australian Research Council | LE0668477 | Rohan A. Davis |
| Australian Research Council | LE140100119 | Rohan A. Davis |
| National Health and Medical Research Council | GNT2017383 | Kate L. Seib |

## AUTHOR CONTRIBUTIONS

Alice Ascari, Data curation, Investigation, Writing – review and editing | Taha, Data curation, Investigation, Writing – review and editing | Rohan A. Davis, Conceptualization, Funding acquisition, Methodology, Project administration, Resources, Writing – review and editing | Kate L. Seib, Conceptualization, Funding acquisition, Project administration, Resources, Supervision, Writing – review and editing | Evgeny A. Semchenko, Conceptualization, Data curation, Formal analysis, Investigation, Methodology, Project administration, Resources, Supervision, Validation, Visualization, Writing – original draft, Writing – review and editing

## ADDITIONAL FILES

The following material is available online.

### Supplemental Material

**Supplemental material (Spectrum02724-25-s0001.docx).** Dataset S1: List of 54 chemically defined natural compounds from NatureBank library. Method S1: UHPLC-MS compound preparation and purity analysis conditions. Figure S1: UHPLC-MS analysis of celastrol. Figure S2: UHPLC-MS analysis of pristimerin. Table S1: MIC and MBC of CL, PR, AZ, and CRO in Ng.

### Open Peer Review

**PEER REVIEW HISTORY (review-history.pdf).** An accounting of the reviewer comments and feedback.

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
