## [Reviewer comments · Microbiology Spectrum]

Microbiology Spectrum

Antimicrobial Activity of Natural Quinone-Methide Triterpenoids Celastrol and Pristimerin Against Pathogenic *Neisseria*

Alice Ascari, Taha Taha, Rohan Davis, Kate Seib, and Evgeny Semchenko

Corresponding Author(s): Evgeny Semchenko, Griffith University - Gold Coast Campus

Review Timeline:

Submission Date:	September 2, 2025
Editorial Decision:	September 25, 2025
Revision Received:	November 14, 2025
Accepted:	December 2, 2025

Editor: Ana-Maria Dragoi

Reviewer(s): Disclosure of reviewer identity is with reference to reviewer comments included in decision letter(s). The following individuals involved in review of your submission have agreed to reveal their identity: William M Shafer (Reviewer #1)

Transaction Report:

DOI: <https://doi.org/10.1128/spectrum.02724-25>

Re: Spectrum02724-25 (Antimicrobial Activity of Natural Quinone Triterpenoids Celastrol and Pristimerin Against Pathogenic Neisseria)

Dear Dr. Evgeny A Semchenko:

Thank you for the privilege of reviewing your work. While both reviewers were enthusiastic about the potential for new gonorrhoea therapies, they noted a few necessary modifications. Below you will find my comments, instructions from the Spectrum editorial office, and the reviewer comments.

Revision Guidelines

Sincerely,
Ana-Maria Dragoi
Editor
Microbiology Spectrum

Reviewer #1 (Comments for the Author):

There is a clear need for new, safe antimicrobials that are effective against *N. gonorrhoeae* (Ng). Towards this goal this manuscript reports on two natural products (celastrol and pristimerin) that have antimicrobial action against Ng and to a lesser extent *N. meningitidis*. Of these, pristimerin appears to be less toxic to cervical epithelial cells and can be considered as a lead compound for future studies to increase antimicrobial potency. While there is clinical significance to the results described some

technical issues and editorial changes are needed to strengthen the paper.

Technical issues: With respect to the antimicrobial testing assay and results presented in Table 1 it is unclear as to why the authors used the microbroth dilution assay as CLSI guidelines indicate that the agar dilution system is the gold standard for testing against Ng. I would accept the results presented in Table 1 if the authors showed that both assays gave identical MIC results. Also, providing results from other publications is not recommended as the studies by Unemo et al. employed agar dilution or Etest.

Editorial issues:

1. Line 194 (subtitle): insert the word "cell" after cervical as this is more precise.
2. line 216: please provide a reference for a value of 10 as a threshold for acceptable SI.
3. line 247: Given the MIC and MBC values of the two compounds, the characterization that the compounds have strong antimicrobial action is misleading especially compared to ceftriaxone.
4. The authors implicate efflux pumps possessed by Ng as having the capacity to recognise the compounds given the impact of loss of MtrE. As this outer membrane protein serves as an export channel for multiple pumps it is important to identify which pump is responsible for presumed efflux. This could be done using transporter mutants such as was done by Golparian et al in their 2014 paper. Moreover, the group could use the self-inhibiting peptides described in ref. 20 to ascertain if the MtrCDE efflux pump is involved.
5. Ref. 43 should be changed to Zaronelli et al. 1999. AAC 43(10):2468-2472 as this is a better description for the ability of the MtrCDE efflux pump to export azithromycin.
6. It is noteworthy that the structures of both compounds resemble to some extent human sex hormones that can be exported by efflux; perhaps the authors could mention this given recent reports by the Brennan lab on the ability of progesterone and testosterone to bind the transcriptional repressor that depresses efflux pump gene expression.

Reviewer #2 (Comments for the Author):

The manuscript submitted by Ascari et al. investigates the bactericidal activity of two natural compounds derived from plants, celastrol and pristimerin, for *Neisseria gonorrhoeae* and *N. meningitidis*. Bactericidal activity was found for 10 gonococcal and 5 meningococcal strains with celastrol exhibiting higher bactericidal activity than pristimerin for both species based on MIC and MBC data. Both compounds also exhibited bactericidal activity for gonococci infecting E6/E7 transformed cervical epithelial cells with celastrol the more active of the two. However, in a cytotoxicity assay using the cervical epithelial cells, celastrol exhibited dose-dependent cytotoxicity whereas pristimerin showed substantially lower cytotoxicity. In general, this study provides evidence in support of new potential therapeutics for gonococcal infection for which there is an urgent need given the significant increase in cases of multidrug-resistant gonorrhea and the lack of a vaccine.

Specific Comments:

1. Introduction, Paragraph 3, lines 69-72; Two novel therapeutics, cell-penetrating peptides and LpxC inhibitors, targeting gonococci are mentioned without providing references for the reader.
2. What was the rationale for screening 54 compounds given the thousands of natural product extracts available from NatureBank?
3. It would be informative to show the results of all 54 compounds or at least provide information on the potency criteria that was used to choose the final 2 compounds out of the pool of 54 test samples.
4. DMSO was used to solubilize the extracted celastrol and pristimerin compounds, which means that there was a level of DMSO in the MIC and MBC assays. However, there is no mention of adding DMSO to the antibiotic containing or control wells.
5. Multiple inconsistencies exist with the MICs listed in Table 1 and those reported in the Evert 2022 and Unemo 2024 papers that are referenced in Table 1. For instance, Table 1 lists the MIC of WHO K for azithromycin and ceftriaxone as 0.5 and 0.064, respectively, however Table 2 in the Evert paper lists the MIC for them as 0.25 and 0.031, respectively. In addition, the MICs referenced in the Unemo paper for 1291 and WHO X Δ mtrE are not in that paper. Further, the MICs for WHO F, WHO H, WHO Q, WHO R, WHO S2, and WHO Z referenced from the Evert paper are not in that paper. Ideally, as with FA1090, determination of the MICs for the other 10 referenced strains would be done by the authors rather than referencing papers to allow for direct comparison of the MICs of celastrol and pristimerin with antibiotics using the same broth microdilution method.
6. It would be useful to have a positive control antibiotic for NM in Table 2 as for GC in Table 1
7. In Figure 2a, in the presence of cervical epithelial cells, there appears to be viable WHO X bacteria with pristimerin treatment as high as approximately 24 ug/ml (10% viability), yet the MBC in Table 1 is given as 5.8 ug/ml. Comment on the impact of cervical cells on the potency of pristimerin.

8. What was the MOI for the E6/E7 cervical cell infection studies presented in Figure 2a?

9. A paper by Nizer et al. (J. Ethnopharmacol. 266:113423, 2021) reported that pristimerin had no bactericidal activity (MIC >400 µg/ml) for nine gram-negative bacteria including *E. coli*, *P. aeruginosa*, and *A. baumannii*. The authors should include this reference and comment on the susceptibility of *Neisseria* to pristimerin compared to other gram-negatives. In addition, Nizer also reported high cytotoxicity of pristimerin for Vero cells (CC50 = 6).

The manuscript submitted by Ascari et al. investigates the bactericidal activity of two natural compounds derived from plants, celastrol and pristimerin, for *Neisseria gonorrhoeae* and *N. meningitidis*. Bactericidal activity was found for 10 gonococcal and 5 meningococcal strains with celastrol exhibiting higher bactericidal activity than pristimerin for both species based on MIC and MBC data. Both compounds also exhibited bactericidal activity for gonococci infecting E6/E7 transformed cervical epithelial cells with celastrol the more active of the two. However, in a cytotoxicity assay using the cervical epithelial cells, celastrol exhibited dose-dependent cytotoxicity whereas pristimerin showed substantially lower cytotoxicity. In general, this study provides evidence in support of new potential therapeutics for gonococcal infection for which there is an urgent need given the significant increase in cases of multidrug-resistant gonorrhea and the lack of a vaccine.

Specific Comments:

1. Introduction, Paragraph 3, lines 69-72; Two novel therapeutics, cell-penetrating peptides and LpxC inhibitors, targeting gonococci are mentioned without providing references for the reader.
2. What was the rationale for screening 54 compounds given the thousands of natural product extracts available from NatureBank?
3. It would be informative to show the results of all 54 compounds or at least provide information on the potency criteria that was used to choose the final 2 compounds out of the pool of 54 test samples.
4. DMSO was used to solubilize the extracted celastrol and pristimerin compounds, which means that there was a level of DMSO in the MIC and MBC assays. However, there is no mention of adding DMSO to the antibiotic containing or control wells.
5. Multiple inconsistencies exist with the MICs listed in Table 1 and those reported in the Evert 2022 and Unemo 2024 papers that are referenced in Table 1. For instance, Table 1 lists the MIC of WHO K for azithromycin and ceftriaxone as 0.5 and 0.064, respectively, however Table 2 in the Evert paper lists the MIC for them as 0.25 and 0.031, respectively. In addition, the MICs referenced in the Unemo paper for 1291 and WHO X Δ mtrE are not in that paper. Further, the MICs for WHO F, WHO H, WHO Q, WHO R, WHO S2, and WHO Z referenced from the Evert paper are not in that paper. Ideally, as with FA1090, determination of the MICs for the other 10 referenced strains would be done by the authors rather than referencing papers to allow for direct comparison of the MICs of celastrol and pristimerin with antibiotics using the same broth microdilution method.
6. It would be useful to have a positive control antibiotic for NM in Table 2 as for GC in Table 1
7. In Figure 2a, in the presence of cervical epithelial cells, there appears to be viable WHO X bacteria with pristimerin treatment as high as approximately 24 ug/ml (10% viability), yet the MBC in Table 1 is given as 5.8 ug/ml. Comment on the impact of cervical cells on the potency of pristimerin.
8. What was the MOI for the E6/E7 cervical cell infection studies presented in Figure 2a?
9. A paper by Nizer et al. (*J. Ethnopharmacol.* 266:113423, 2021) reported that pristimerin had no bactericidal activity (MIC >400 μ g/ml) for nine gram-negative bacteria including *E. coli*, *P. aeruginosa*, and *A. baumannii*. The authors should include this reference and comment on the susceptibility of *Neisseria* to pristimerin compared to other gram-negatives. In addition, Nizer also reported high cytotoxicity of pristimerin for Vero cells (CC₅₀ = 6).

REVIEWER #1 - COMMENTS FOR THE AUTHOR

There is a clear need for new, safe antimicrobials that are effective against *N. gonorrhoeae* (Ng). Towards this goal this manuscript reports on two natural products (celastrol and pristimerin) that have antimicrobial action against Ng and to a lesser extent *N. meningitidis*. Of these, pristimerin appears to be less toxic to cervical epithelial cells and can be considered as a lead compound for future studies to increase antimicrobial potency. While there is clinical significance to the results described some technical issues and editorial changes are needed to strengthen the paper.

Technical issues: With respect to the antimicrobial testing assay and results presented in Table 1 it is unclear as to why the authors used the microbroth dilution assay as CLSI guidelines indicate that the agar dilution system is the gold standard for testing against Ng. I would accept the results presented in Table 1 if the authors showed that both assays gave identical MIC results. Also, providing results from other publications is not recommended as the studies by Unemo et al. employed agar dilution or Etest.

Response:

We used the broth microdilution method to enable higher throughput and to maintain consistency with subsequent infection-model experiments. We fully acknowledge that agar dilution remains the CLSI-recommended gold standard for antimicrobial susceptibility testing of Ng. Nonetheless, the use of broth microdilution has been explored for over 30 years, with studies such as Shapiro (1984) and Geer (1989) demonstrating comparable MIC values to agar dilution method. Importantly, our goal was to assess relative strain sensitivity rather than to establish definitive MIC breakpoints or conduct cross-method comparisons.

In addition, and, to improve clarity, we have revised Table 1 to present only celastrol and pristimerin MIC/MBC values (generated by broth microdilution), which are central to the study's conclusions. However, we retained azithromycin and ceftriaxone sensitivity profiles for the strain panel, as this information is important to one of our key observations that Ng sensitivity to celastrol and pristimerin does not correlate with sensitivity to these clinically used antibiotics. To ensure clarity, the MIC values for azithromycin and ceftriaxone (against strains 1291, WHO X, and WHO X $\Delta mtrE$, as determined in this study) have been moved to a new supplementary table ("Supplementary Table S1"). This separation avoids potential cross-method comparisons and ensures the main table remains methodologically consistent.

EDITORIAL ISSUES

1. Line 194 (subtitle): insert the word "cell" after cervical as this is more precise.

This has been amended as per the reviewer's suggestion. The subtitle now reads "*In vitro* efficacy of compounds against cervical cell infection by *N. gonorrhoeae*."

2. Line 216: please provide a reference for a value of 10 as a threshold for acceptable SI.

This has been amended, and we now cite a standard medicinal chemistry/natural products benchmarking reference for $SI \geq 10$ and clarify that SI thresholds are context dependent. The sentence has been revised to: "*Although this value could not be precisely quantified due to the upper limit of the cytotoxicity assay, it exceeds the commonly applied threshold of 10 for preliminary selectivity in cell-based antibacterial discovery (34, 35), suggesting a favourable safety margin and supporting its potential for further development.*" (lines 236-239).

3. Line 247: Given the MIC and MBC values of the two compounds, the characterization that the compounds have strong antimicrobial action is misleading especially compared to ceftriaxone.

We have amended this sentence which now reads “*We subsequently showed that both celastrol and pristimerin exhibit antimicrobial activity, with MICs and MBCs indicating both bacteriostatic and bactericidal effects against diverse Ng and Nm strains*” (lines 274-276). Additionally, we have updated wording in the abstract to reflect the same (lines 36-38).

4. The authors implicate efflux pumps possessed by Ng as having the capacity to recognise the compounds given the impact of loss of MtrE. As this outer membrane protein serves as an export channel for multiple pumps it is important to identify which pump is responsible for presumed efflux. This could be done using transporter mutants such as was done by Golparian et al in their 2014 paper. Moreover, the group could use the self-inhibiting peptides described in ref. 20 to ascertain if the MtrCDE efflux pump is involved.

We agree that identifying the specific efflux system responsible is important however it is not within the scope of this work. We have clarified that the increased susceptibility observed in the $\Delta mtrE$ mutant is consistent with RND-type efflux involvement and does not conclusively assign pump specificity. Our revised the manuscript now reads “*Additionally, the Resistance-Nodulation-Division (RND) type efflux pump outer membrane channel MtrE mutant strain WHO X $\Delta mtrE$ was two-four times more susceptible to both compounds than the WHO X wildtype strain (Table 1).*” Lines 199-201. Additionally, we have framed this as a key direction for future work, citing Golparian et al. (2014). Revised text now reads “*These shared physicochemical features suggest a potential intersection with Ng efflux regulation, warranting future investigation into their binding to MtrR and activity across isogenic transporter mutants (48, 50)*” (lines 301-304).

5. Ref. 43 should be changed to Zarattonelli et al. 1999. AAC 43(10):2468-2472 as this is a better description for the ability of the MtrCDE efflux pump to export azithromycin.

This reference has been updated.

6. It is noteworthy that the structures of both compounds resemble to some extent human sex hormones that can be exported by efflux; perhaps the authors could mention this given recent reports by the Brennan lab on the ability of progesterone and testosterone to bind the transcriptional repressor that depresses efflux pump gene expression.

These structures do share some similarities, and we have revised our discussion to further explore this and present it as a direction for future investigation beyond the scope of the current study. Our revised Discussion now reads “*Interestingly, the WHO X MtrE efflux pump mutant strain was more susceptible to both celastrol and pristimerin, suggesting that Ng may detoxify these compounds via MtrE-dependent efflux, which preferentially exports hydrophobic and amphiphilic molecules such as azithromycin (49, 50). Notably, celastrol and pristimerin exhibit pronounced hydrophobicity and possess planar, conjugated ring systems that may also facilitate interaction with the efflux regulator MtrR through π - π and general hydrophobic contacts, akin to steroid hormones such as progesterone and testosterone, which have been shown to bind MtrR and modulate mtrCDE expression (51). These shared physicochemical features suggest a potential intersection with Ng efflux regulation, warranting future investigation into their binding to MtrR and activity across isogenic transporter mutants (48, 51). While the efflux interaction remains hypothetical, celastrol and pristimerin nevertheless demonstrated robust activity against the WHO panel*

of Ng strains with extensive antimicrobial resistance profiles, including those with reduced susceptibility to ceftriaxone and high-level resistance to azithromycin, highlighting the therapeutic potential of triterpenoid scaffolds in the context of multidrug- and extensively drug-resistant gonococcal infections.” Lines 294-308.

REVIEWER #2 - COMMENTS FOR THE AUTHOR

The manuscript submitted by Ascari et al. investigates the bactericidal activity of two natural compounds derived from plants, celastrol and pristimerin, for *Neisseria gonorrhoeae* and *N. meningitidis*. Bactericidal activity was found for 10 gonococcal and 5 meningococcal strains with celastrol exhibiting higher bactericidal activity than pristimerin for both species based on MIC and MBC data. Both compounds also exhibited bactericidal activity for gonococci infecting E6/E7 transformed cervical epithelial cells with celastrol the more active of the two. However, in a cytotoxicity assay using the cervical epithelial cells, celastrol exhibited dose-dependent cytotoxicity whereas pristimerin showed substantially lower cytotoxicity. In general, this study provides evidence in support of new potential therapeutics for gonococcal infection for which there is an urgent need given the significant increase in cases of multidrug-resistant gonorrhoea and the lack of a vaccine.

SPECIFIC COMMENTS:

1. Introduction, Paragraph 3, lines 69-72; Two novel therapeutics, cell-penetrating peptides and LpxC inhibitors, targeting gonococci are mentioned without providing references for the reader.

These references have been added.

2. What was the rationale for screening 54 compounds given the thousands of natural product extracts available from NatureBank?

To clarify our rationale for screening a library of compounds vs natural product extracts, we have amended the first paragraph of the Discussion section, which now reads “*Nature has historically been a cornerstone of drug discovery and continues to hold immense promise for identifying novel bioactive compounds. NatureBank, a unique drug discovery platform, harnesses this potential through its extensive library of natural product extracts (~21,000), fractions (~105,000) and compounds (~54) derived from Australian plants, fungi, and marine invertebrates (23). However, natural product extracts are complex mixtures, which are not chemically defined, and their constituents are often present in very low concentrations, which could be below the threshold required to elicit a measurable biological activity. Thus, our approach to identifying novel antimicrobials targeting pathogenic Neisseria was to focus on a small library of high quality chemically defined compounds.*” Lines 259-267.

The Materials and Methods section now also reads “*A library comprised of 54 high quality, chemically defined natural compounds was obtained from NatureBank (Dataset S1).*” Lines 111-112.

3. It would be informative to show the results of all 54 compounds or at least provide information on the potency criteria that was used to choose the final 2 compounds out of the pool of 54 test samples.

As described in the Results section, we screened a library of 54 compounds from NatureBank using a high-throughput growth inhibition assay against *N. gonorrhoeae* strain

1291, with each compound tested at a single concentration of 200 μ M. Of these, only two structurally related compounds (celestrol and pristimerin) showed clear inhibition of bacterial growth and were selected for further investigation (Fig. 1). Compounds that did not inhibit growth at this concentration were excluded from further analysis, but we have now included the full list of 54 compounds in supplementary Dataset S1. The following statement was added to the *Results* section for clarity “*The potency criterion used for selection was based on complete inhibition of visible bacterial growth at the tested concentration, as determined by visual scoring in accordance with CLSI guidelines.*” Lines 186-188.

4. DMSO was used to solubilize the extracted celestrol and pristimerin compounds, which means that there was a level of DMSO in the MIC and MBC assays. However, there is no mention of adding DMSO to the antibiotic containing or control wells.

We have clarified in the revised *Materials and Methods* section that the DMSO concentration was matched across all wells, including those containing ‘no-compound’ controls. Specifically, control wells were supplemented with the same final concentration of DMSO as the test wells. We also note that DMSO had no observable effect on bacterial growth or viability in control wells. These sections now read; “*Control wells contained media alone and media supplemented with a matching concentration of DMSO (2%), which did not have any visible impact on bacterial growth.*” (lines 114-116) and “*All wells, including azithromycin and no-compound controls, contained matched DMSO ($\leq 0.5\%$ v/v), which had no effect on bacterial growth or viability.*” Lines 133-135.

5. Multiple inconsistencies exist with the MICs listed in Table 1 and those reported in the Evert 2022 and Unemo 2024 papers that are referenced in Table 1. For instance, Table 1 lists the MIC of WHO K for azithromycin and ceftriaxone as 0.5 and 0.064, respectively, however Table 2 in the Evert paper lists the MIC for them as 0.25 and 0.031, respectively. In addition, the MICs referenced in the Unemo paper for 1291 and WHO X Δ mtrE are not in that paper. Further, the MICs for WHO F, WHO H, WHO Q, WHO R, WHO S2, and WHO Z referenced from the Evert paper are not in that paper. Ideally, as with FA1090, determination of the MICs for the other 10 referenced strains would be done by the authors rather than referencing papers to allow for direct comparison of the MICs of celestrol and pristimerin with antibiotics using the same broth microdilution method.

We have address this issue by making the following changes: (i) Table 1 now reports only our celestrol/pristimerin MIC/MBC values; (ii) we removed literature antibiotic MIC values from the main table to avoid cross-method confusion; (iii) we created “Supplementary Table S1” listing MICs for azithromycin, ceftriaxone, celestrol and pristimerin for Ng 1291, WHO X and WHO X Δ mtrE strains, which were determined using broth microdilution method in this study.

6. It would be useful to have a positive control antibiotic for NM in Table 2 as for GC in Table 1.

Table 1 has been revised as per reviewer 1 and 2, and no longer reports azithromycin or ceftriaxone control data for Ng. To maintain consistency across all tables, and to avoid redundancy, we have not added these values to Table 2. Note that azithromycin was used as a positive control, which is described in *Materials and Methods* section lines 133-134.

7. In Figure 2a, in the presence of cervical epithelial cells, there appears to be viable WHO X bacteria with pristimerin treatment as high as approximately 24 ug/ml (10% viability), yet the MBC in Table 1 is given as 5.8 ug/ml. Comment on the impact of cervical cells on the potency of pristimerin.

We have added following to the Discussion section to address this point: “*Notably, higher bactericidal concentrations of compounds were required in the cervical cell infection model compared to planktonic MIC/MBC values. This reduced susceptibility is a well-established phenomenon in cell-based infection assays and likely stems from host-cell environmental factors, including bacterial adhesion, invasion, and intracellular localization, which are known to shift concentration-response curves.*” Lines 327-332.

8. What was the MOI for the E6/E7 cervical cell infection studies presented in Figure 2a?

We have now reported the MOI in *Materials and Methods* section. Lines 152-154 now read “*Assay wells were inoculated with 50 μ L of Ng suspension ($\sim 10^7$ colony forming units (CFU)/mL; multiplicity of infection (MOI):10) and plates incubated for 1 hour at 37 °C with 5% CO₂.*”

9. A paper by Nizer et al. (J. Ethnopharmacol. 266:113423, 2021) reported that pristimerin had no bactericidal activity (MIC >400 μ g/ml) for nine gram-negative bacteria including *E. coli*, *P. aeruginosa*, and *A. baumannii*. The authors should include this reference and comment on the susceptibility of *Neisseria* to pristimerin compared to other gram-negatives. In addition, Nizer also reported high cytotoxicity of pristimerin for Vero cells (CC50 = 6).

We have added two sections addressing these points in the *Discussion*, which read “*In contrast to several non-Neisseria Gram-negative species where pristimerin showed no bactericidal activity at concentrations exceeding 400 mg/L (47), Neisseria species exhibited significantly lower MICs under our assay conditions, indicating greater susceptibility. This suggests that species-specific envelope or physiological features may influence pristimerin activity (48).*” (Lines 290-294) and “*Pristimerin has also been reported to exhibit variable cytotoxicity (e.g., CC50 \sim 6 mg/L in Vero cells) (47), underscoring differences across cell types and assay conditions.*” Lines 336-338.

Re: Spectrum02724-25R1 (Antimicrobial Activity of Natural Quinone-Methide Triterpenoids Celastrol and Pristimerin Against Pathogenic *Neisseria*)

Dear Dr. Evgeny A Semchenko:

Your manuscript has been accepted, and I am forwarding it to the ASM production staff for publication. Your paper will first be checked to make sure all elements meet the technical requirements. ASM staff will contact you if anything needs to be revised before copyediting and production can begin. Otherwise, you will be notified when your proofs are ready to be viewed.

Sincerely,
Ana-Maria Dragoi
Editor
Microbiology Spectrum

Reviewer #1 (Comments for the Author):

The authors have addressed all of my concerns and modified their manuscript accordingly.